# Triple Network Resting State Connectivity Predicts Distress Tolerance and Is Associated with Cocaine Use

**DOI:** 10.3390/jcm8122135

**Published:** 2019-12-03

**Authors:** Elizabeth D. Reese, Jennifer Y. Yi, Katlyn G. McKay, Elliot A. Stein, Thomas J. Ross, Stacey B. Daughters

**Affiliations:** 1Department of Psychology and Neuroscience, University of North Carolina, Chapel Hill, NC 27514, USA; edreese@email.unc.edu (E.D.R.); yijy@email.unc.edu (J.Y.Y.); katlyn.g.mckay@vanderbilt.edu (K.G.M.); 2Neuroimaging Research Branch, Intramural Research Program, National Institute on Drug Abuse, National Institutes of Health, Baltimore, MD 21224, USA; estein@mail.nih.gov (E.A.S.); tross@mail.nih.gov (T.J.R.)

**Keywords:** distress tolerance, resting state functional connectivity, substance use disorder, cocaine, salience, default mode, executive control

## Abstract

Distress tolerance (DT), a predictor of substance use treatment retention and post-treatment relapse, is associated with task based neural activation in regions located within the salience (SN), default mode (DMN), and executive control networks (ECN). The impact of network connectivity on DT has yet to be investigated. The aim of the present study was to test within and between network resting-state functional connectivity (rsFC) associations with DT, and the impact of cocaine use on this relationship. Twenty-nine adults reporting regular cocaine use (CU) and 28 matched healthy control individuals (HC), underwent resting-state functional magnetic resonance imaging followed by the completion of two counterbalanced, computerized DT tasks. Dual-regression analysis was used to derive within and between network rsFC of the SN, DMN, and lateralized (left and right) ECN. Cox proportional-hazards survival models were used to test the interactive effect of rsFC and group on DT. The association between cocaine use severity, rsFC, and DT was tested within the CU group. Lower LECN and higher DMN-SN rsFC were associated with DT impairment. Greater amount of cocaine use per using day was associated with greater DMN-SN rsFC. The findings emphasize the role of neural resource allocation within the ECN and between DMN-SN on distress tolerance.

## 1. Introduction

Distress tolerance (DT), defined as the ability to persist in goal directed behavior while experiencing psychological and physiological distress, is an established correlate of substance use disorders (SUD) and a risk factor for poor substance use treatment response. Lower DT is associated with a greater frequency of substance use [1], early substance use treatment dropout [2,3,4], and higher rates of post treatment relapse to substance use [5,6,7,8]. In addition, longer abstinence duration following substance use treatment entry is associated with greater improvement in DT over time, whereas greater frequency of post treatment substance use predicts attenuations in DT improvement [9].

DT is conceptualized within negative reinforcement models of substance use [10,11] so that lower DT increases the risk of drug seeking and taking via the increased desire to reduce negative affective and physiological symptoms of withdrawal, thereby negatively reinforcing reliance on immediate reward over more adaptive, goal-directed behavior (e.g., abstinence). Although consistent behavioral evidence for a link between DT and indices of SUD and treatment response exists, the neurobiological mechanisms contributing to negative reinforcement driven behavior and DT have not been established. There is growing evidence that neural network activity in the task-free (i.e., resting state) condition are linked to addiction [12,13,14], with an emphasis on neural network connectivity within and between three large-scale neural networks (i.e., default mode network [DMN], executive control network [ECN], and salience network [SN]), which are critical for adaptive emotional and cognitive processing of relevant information. The DMN is responsible for the processing of internally relevant stimuli, self-referential thinking, and perspective-taking [15], the ECN is central to the processing of externally oriented stimuli [16,17], and the SN is implicated in the modulation of activity in both the DMN and ECN via the allocation of attentional resources based on the initial processing of salient information [17,18].

Neural network connectivity between these networks is implicated in addiction, such that deficient modulation of the DMN by the SN is thought to result in biased attentional allocation toward internally oriented stimuli (e.g., withdrawal symptoms) and away from externally oriented information processing, critical for the execution of goal-directed behavior [13,14]. Empirical support is accumulating, with associations reported for within and between DMN, SN, and ECN resting state functional connectivity (rsFC) with both the presence of a SUD and post substance use treatment relapse. Adults with SUD evidence increased DMN and reduced ECN and SN resting state connectivity (rsFC) relative to healthy controls [19,20,21]. Among adults with alcohol use disorder, greater substance use severity is associated with reduced ECN rsFC, and a higher likelihood of relapse is associated with reduced ECN and SN rsFC [22]. Adults with cocaine dependence who relapse within 30-days post treatment demonstrate weaker left SN-ECN and strong interhemispheric ECN rsFC compared to those who remain abstinent [23]. Finally, stronger DMN-SN and weaker ECN-DMN rsFC has been found among smokers under acute abstinence [24] and is associated with relapse within 3-months among heroin dependent men [25]. 

An investigation into the neural indices of DT corroborates the importance of these three neural networks. Among adults with cocaine use disorder, DT is significantly associated with neural activation during a DT task in regions that comprise primary nodes of the ECN, SN, and DMN. In addition, DT is positively associated with connectivity between nodes of the ECN and SN, namely the dorsolateral prefrontal cortex (dlPFC) and ventromedial prefrontal cortex/anterior cingulate cortex (vmPFC/ACC) [26], suggesting that coordinated activity between regions involved in salience detection (SN) and executive control (ECN) may be critical for the ability to persist in goal-directed activity in the face of affective distress. Moreover, it may be that biased attentional processing toward internally relevant cues observed in SUD is a result of altered DMN-SN connectivity at the expense of allocating resources toward top-down ECN processes critical to goal-directed behavior (e.g., abstinence), particularly when experiencing affective distress. However, interactions within and between resting-state networks, rather than regions of interest within individual networks, have yet to be tested in relation to DT. This is important to evaluate given the emphasis of large-scale network connectivity in neurobiological models of SUD [13,14]. We hypothesize that dysregulated connectivity patterns within and between these networks contribute to deficits in DT and subsequent increased risk for substance use disorder.

The primary aim of the current study was to test the association between DT and rsFC within and between the DMN, SN, and ECN in adults who regularly use cocaine (CU) and a healthy control (HC) comparison group. It was predicted that higher DT would be associated with reduced FC strength within the DMN, higher connectivity within the ECN and SN, and stronger between network connectivity in ECN-SN relative to DMN-SN, with CU individuals hypothesized to evidence stronger associations between rsFC and DT when compared to HC. Two exploratory aims tested the unique influence of within and between network rsFC on DT, and within the CU group, the association between cocaine use severity with rsFC and DT.

## 2. Method

### 2.1. Participants

Study procedures were approved by the National Institute on Drug Abuse Institutional Review Board (10-DA-N453; 5/1/09). Sixty-three participants were recruited from the general population of a mid-Atlantic metropolitan area in the United States. All participants were right-handed and between 18 and 55 years of age. Healthy control (HC) participants were non-smokers and did not meet criteria for any current or past substance abuse or dependence as assessed by the Structured Clinical Interview for DSM-IV-TR (SCID-I/NP; [27]) or endorse any use of illicit substances in the past 30 days. Cocaine use (CU) participants endorsed regular past year cocaine use (i.e., ≥two times per week), and did not meet DSM-IV criteria for current or past substance dependence for any substance other than cocaine or nicotine. Exclusion criteria for all participants included: (1) functional magnetic resonance imaging (fMRI) contraindications (e.g., pregnancy, implanted metallic devices, claustrophobia, neurological illness, cardiovascular disease); (2) current DSM-IV Axis-I disorders; (3) acute drug intoxication or positive urinalysis at pre-scan; and (4) cognitive impairment (i.e., Intelligence Quotient [IQ] < 85). Six individuals were excluded from study analyses due to claustrophobia (*n* = 1 CU), study drop-out prior to completion of pre-scan screen (*n* = 3 CU), excessive head motion during the resting state scan (*n* = 1 HC), and self-reported abstinence from cocaine use despite endorsing recent use at screen (*n* = 1 CU). The final sample included 29 CU (*M_age_* = 41.31 ± 8.04, 93% male, 79% Black/African American, 17% Caucasian, 3% Other *M_IQ_* = 101.56 ± 11.88) and 28 HC (*M_age_* = 39.57 ± 8.90, 61% male, 75% Black/African American, 18% Caucasian, 4% Hispanic, *M_IQ_* = 106.29 ± 12.75). Mean and standard deviation statistics are reported here and elsewhere as M±SD unless otherwise indicated. 

### 2.2. General Procedure

All participants completed a pre-scan screen, which included participant informed consent, a urine screen and breathalyzer test for current drug and alcohol use, respectively. All CU participants were given the opportunity to smoke a cigarette approximately 60 minutes prior to entering the MRI scanner. The MRI scan session was part of a larger study and included, in the following order: structural scan, two counterbalanced cognitive tasks, a 7-minute resting state scan, and two counterbalanced DT tasks. After the scan, participants completed self-report measures and were debriefed and compensated. 

### 2.3. Measures

#### 2.3.1. Demographic Information and Sample Characteristics

Participants self-reported sociodemographic information including age, gender, and race; symptoms of nicotine dependence, Fägerstrom Test for Nicotine Dependence (FTND; [28]); and depression and anxiety symptoms, Beck Depression Inventory–II (BDI-II; [29]) and the Beck Anxiety Inventory (BAI; [30]). IQ was measured with the clinician administered Wechsler Abbreviated Scale of Intelligence (WASI-II; [31]). Daily past 30-day cocaine use was self-reported via the Timeline Follow-Back (TLFB; [32]). Two variables were calculated to reflect cocaine use severity. Cocaine use frequency was calculated as the ratio of cocaine use days to number of days assessed. Amount of cocaine use per using day was calculated as the total grams used divided by the number of days used.

#### 2.3.2. Distress Tolerance

Paced Auditory Serial Addition Task for fMRI (PASAT-M; [26]). The PASAT-M is a 10-minute block design task consisting of four phases: easy, latency, distress, and distress tolerance (DT). A series of numbers flash on the screen one at a time and the participant is instructed to add the current number on the screen to the previously presented number, using an MRI compatible joystick to indicate the correct answer from four possible choices before the subsequent number appears (Appendix A). Throughout each phase, correct responses result in a pleasant bell sound and a one-point increase in the participants’ score. During the latency and hard phases, incorrect and/or slow responses result in an explosion noise and a one-point decrease in points. Task difficulty (i.e., time between number presentations) is titrated to participant skill level during the 5-minute latency phase. During the final two phases, the latency between number presentations is set to 2.5× greater than the participant’s calculated skill level, resulting in forced failure. During the DT phase, participants are instructed that they can win back points and no longer lose points for incorrect responses, yet are given the option to quit the task at any time until the task terminates on its own after 10 minutes. Participants are incentivized to persist for the duration of the DT phase as they are told that the monetary compensation they will receive is contingent upon task performance across all phases of the task. Task performance on this task is defined as the latency between number presentations as determined during the latency phase. 

Mirror Tracing Persistence Task for fMRI (MTPT-M). The Mirror Tracing Persistence Task for fMRI (MTPT-M) was adapted from the computerized version of the Mirror Tracing Persistence Task (MTPT-C; [33]). A series of star figures was presented on the computer screen one at a time and participants were instructed to trace a red dot along the tapered edge of the star using an fMRI compatible joystick programmed to move the dot in the opposite direction (Appendix A). Errors occur when the red dot is moved too slowly or tracks outside of the star’s boundary, both of which result in aversive auditory feedback (i.e., buzzer sound) and require participants to restart tracing from the beginning. Like the PASAT-M, the task is a block design with four phases (i.e., easy, latency, distress, and DT), and difficulty is titrated to participant skill level during the latency phase and applied during the final two phases. Difficulty is adjusted by tapering the width of the star edge and adjusting the speed of the red dot. During the final DT phase, participants are told they will no longer lose points for errors and can only win back points. Participants are given the option of quitting the task at any time until the task terminates on its own after 10 minutes, and similar to the PASAT-M, participants are incentivized to persist for the duration of the DT phase as they are told that the monetary compensation they will receive is contingent upon task performance across all phases of the task. Task performance on the MTPT-M is defined as the mean cursor speed (i.e., red dot) during the latency phase. 

Task data. DT was calculated as time in seconds until the participant quit a DT task. For instance, if a participant persisted for the entire 600 seconds (i.e., 10 minutes) of Task 1 without quitting, and then quit 45 seconds into the second task (Task 2) then DT = 645 seconds. If a participant quit 35 seconds into the first DT task (Task 1) then DT = 35 seconds, regardless of persistence time on the second task. A composite measure of distress was calculated prior to each task phase (i.e., pre-easy, pre-latency, pre-hard, and pre-DT) as the mean self-reported anxiety, frustration, irritability, and stress on a scale from 0 (none) to 100 (extreme). Motivation to perform well on each task was self-reported at the end of the testing session prior to knowledge about performance/monetary compensation using a 10-point Likert scale ranging from 1 = “not at all motivated” to 10 = “extremely motivated”. Task performance indices for the PASAT-M and MTPT-M as described above were standardized and then categorized into Task 1 and Task 2. 

### 2.4. fMRI Data Acquisition and Analysis

#### 2.4.1. Image Acquisition

Resting state fMRI scans were acquired on a Siemens 3-T Magnetom Trio (Erlangen, Germany) equipped with a 12-channel head coil in one run of 210 volumes utilizing a blipped, gradient echo, echo-planar sequence (TR/TE = 2000/27 ms, flip angle = 78°, 4 mm slice thickness, 22 cm field of view (FOV), 64 × 64 matrix, voxel size = 3.4 × 3.4 × 4 mm^3^, no gap). Thirty-nine oblique, axial slices were obtained 30° to the AC–PC plane [34]. Structural images were acquired utilizing a T1-weighted magnetization-prepared rapid gradient echo (MPRAGE) (TR/TE = 1900/3.5 ms, flip angle = 9˚, 208 slices, 1 mm slice thickness, 26 cm FOV, 256 × 192 matrix, voxel size=1 × 1 mm^2^, no gap). 

#### 2.4.2. Preprocessing

The functional and anatomical data were pre-processed and analyzed using FMRIB’s Software Library (FSL v5.0.9; http://www.fmrib.ox.ac.uk/fsl) using FSL FEAT v. 6.00. Preprocessing of the resting-state fMRI data included motion correction using MCFLIRT, non-brain removal using BET, spatial smoothing using a full-width at half-maximum Gaussian kernel of 5 mm, and high-pass temporal filtering at 100 s. Functional images were co-registered to the individual’s corresponding T1-weighted anatomical image and transformed to standard anatomical space (Montreal Neurological Institute; voxel dimensions 2 × 2 × 2 mm^3^) using FLIRT. In accordance with best practice suggestions to adequately address the effects of motion-related noise in functional imaging data [35], nuisance regressors were extracted from the data including (1) six rotation and translation motion realignment parameters (x, y, z, roll, pitch, yaw) estimated using MCFLIRT during preprocessing, and (2) high-motion volumes identified through FSL’s motion outlier tool using framewise displacement (FD) a weighted metric representing the average of rotation and translation parameter differences. The FD threshold was set to a boxplot cut off of the 75^th^ percentile + 1.5 × the interquartile range. Standardized, nuisance regressed 4-D resting state functional data for each subject were created following the aforementioned preprocessing steps and used in subsequent connectivity analyses. 

#### 2.4.3. Resting State Functional Connectivity (rsFC)

Template maps of resting state networks corresponding to the DMN, SN, and lateralized ECN were derived from independent component analysis of 27 healthy adults [36]. Network template maps contained regions of interest (ROIs) representing primary nodes of the four networks of interest (see [36] for full list of ROIs within each network): lateralized executive control networks (LECN and RECN), the anterior and posterior salience networks (combined to create the SN), and dorsal and ventral default mode networks (combined to create the DMN). The left and right ECN were analyzed separately given the lateralized relationship between the ECN and relapse risk factors including DT (e.g., see [24,25,26]). Subject specific 4-D rsFC data were regressed onto template maps to obtain participant specific time series data corresponding to each network of interest. As censoring of high-motion volumes occurred prior to regressing participant 4D data onto template maps (i.e., in preprocessing steps), these volumes were replaced in participant-specific time series data using a k-nearest-neighbor imputation where the time series data from the immediately previous volume was replaced in the dataset for each censored volume. This method was used to assure temporal continuity in the time-series data.

#### 2.4.4. Within Network Connectivity

Within network connectivity was calculated by regressing subject specific rsFC data onto the ROIs within each network template of interest (see [36] for detailed descriptions of ROIs within each network). This resulted in mean times series data for each ROI of a given network (e.g., SN consisted of 19 ROIs and thus time series data for each), which were next entered into FSLnets to yield a matrix of partial Pearson product–moment correlation coefficients for each subject, representing the correlation between all ROI pairs for a specified network while controlling for correlations between all other ROI pairs. Fisher’s r-to-z transformation was performed on partial correlation coefficients that were then averaged to yield a final within network connectivity value for each network of interest for each subject, which was entered into SPSS for further analyses.

#### 2.4.5. Between Network Connectivity

To calculate between network connectivity, template maps of the SN, DMN, and lateralized ECN were regressed onto subject 4D rsFC data to yield subject-specific time series data for each network, which were then entered into FSLNets. This step yielded a 4 × 4 correlation matrix for each subject consisting of partial Pearson product–moment correlation coefficients representing the direct connection strength, or between network connectivity, for each pair of pre-defined component networks while controlling for all other possible network connections. The resulting partial correlation coefficients representing DMN–SN, LECN–SN, RECN–SN, LECN–DMN, and RECN–DMN coupling were extracted for each participant and entered into SPSS. In accordance with the procedures reported in Lerman et al. [24], the resource allocation index (RAI) for each participant was calculated to reflect the connectivity strength of the SN with both the DMN and ECN. Larger RAI values represent stronger SN coupling with ECN and/or DMN, and values closer to 0 are indicative of weaker coupling between the SN and ECN/DMN. Fisher’s r-to-z transformations were performed on all between network connectivity values including RAI, prior to analyses. 

### 2.5. Data Analysis

The primary dependent variable in the current study was DT: time in seconds until quit. DT task analyses were first conducted to examine group differences in task performance and self-reported motivation to perform each DT task as well as to confirm that the tasks induced distress. Next, potential covariates including baseline sociodemographic characteristics, depression and anxiety symptoms, DT task order, and task motivation and performance were tested via omnibus Cox proportional hazards survival regression for their associations with DT. Variables demonstrating significant associations with DT were included as covariates in subsequent analyses. Cox proportional hazards survival regression, which tests the effects of continuous variables (i.e., within or between network rsFC) on event-based outcomes (i.e., quit DT task, yes/no) over a specified time period (i.e., time in seconds until quit) was used to test all study aims. Odds ratios from uncorrected models were interpreted to evaluate significant effects on DT. Initial study analyses included gender and BDI-II as covariates based on significant group differences. Neither gender nor BDI were significantly associated with the dependent variable in the rsFC models and thus were removed.

#### 2.5.1. Primary Aim: rsFC, Group, and DT

The interactive effect of network rsFC and group on DT was estimated in separate models for each of the within and between network rsFC variables. For each model, covariates, group, and rsFC were entered into Step 1. The rsFC × Group interaction term was entered in Step 2. Continuous predictors were transformed to z scores to facilitate interpretation. Significant RAI models were subjected to post-hoc analyses to delineate the specific between network rsFC variable impacting DT. 

#### 2.5.2. Exploratory Aims

The Cox proportional hazards survival regression model testing the unique contribution of within and between network rsFC predicting DT included significant covariates, group, and significant within and between network rsFC predictors (as determined in the primary aim analyses) in a single step. A Cox proportional hazards survival regression and linear regression models tested the association between cocaine use severity (both cocaine use frequency and amount of cocaine use per using day) within the CU group on DT and rsFC, respectively. The cocaine use per using day variable was positively skewed and therefore, log transformed in all models.

## 3. Results

### 3.1. Group Differences in Participant Characteristics and Resting State Functional Connectivity

Descriptive data and group differences in participant characteristics are reported in Table 1. CU reported past 30-day use of crack/cocaine (*M_days_* = 6.36 ± 6.32), alcohol (*M_days_* = 4.07 ± 5.34), and marijuana (*M_days_* = 0.14 ± 0.59), an average of 4.48 ± 5.39 days of abstinence prior to the scan assessment, and low-to-moderate nicotine dependence (*M_FTND_* = 2.77 ± 2.14). Participants in the HC group reported past 30-day use of alcohol (*M_days_* = 2.12 ± 2.41) and an average of 6.00 ± 5.63 days of abstinence from alcohol prior to the scan assessment. Groups did not differ in age, race, IQ, or anxiety symptoms. There were significantly more females in the HC group, and the CU group reported significantly higher levels of depressive symptoms. The HC group evidenced significantly greater LECN and RECN-SN rsFC, while CU evidenced significantly greater DMN and LECN-SN rsFC. 

### 3.2. Distress Tolerance Task Effects and Identification of Covariates

The DT task effects including descriptive statistics and analyses of between group effects are summarized in Table 2. Analyses were conducted using ANCOVA and univariate general linear model (GLM), controlling for task order. There were no significant group differences in performance or motivation on either task. There was a significant main effect of time on self-reported distress for both tasks, with an increase from the pre-easy to pre-DT phase, with no significant effect of group or a group by time interaction. Sociodemographic characteristics, anxiety symptoms, depressive symptoms, nor DT task effects were associated with DT (Appendix A). 

### 3.3. Within Network rsFC and DT

Cox proportional hazards regression models tested the effect of within network rsFC, group, and the rsFC × Group interaction on DT (Appendix A). The RECN, DMN, and SN models did not yield significant effects. Step 1 of the LECN model was significant (χ^2^(2) = 8.53, p = 0.01), with a significant effect of LECN rsFC on DT (Hazard Ratio = 0.63; p = 0.005; 95% Confidence Interval = 0.45–0.87), so that greater LECN rsFC predicts higher DT (Figure 1a). The odds ratios revealed that a one unit increase in LECN rsFC resulted in a 37.4% decrease in the likelihood of quitting the DT task. The addition of the interaction term in Step 2 did not significantly improve model fit. 

### 3.4. Between Network rSFC and Distress Tolerance

Cox proportional hazards regression models tested the effect of between network rsFC, group, and their interaction on DT (Appendix A). The R.RAI model did not yield significant effects. Step 1 of the L.RAI model was significant (χ^2^(2) = 6.36, p = 0.04), with greater L.RAI significantly predicting higher DT (HR = 0.66; p = 0.01; 95% CI = 0.48–0.91) where a one unit increase in L.RAI resulted in a 44% decrease in the likelihood of quitting the DT task. The inclusion of the interaction term in Step 2 did not significantly improve the model fit. A test of the networks underlying the L.RAI effect was conducted with models including the LECN–DMN, DMN–SN, and LECN–SN (Appendix A). The LECN–SN and LECN–DMN models did not yield significant effects. The DMN–SN model was significant (χ^2^(2) = 6.53, p = 0.04), with greater DMN–SN rsFC significantly predicting lower DT (HR = 1.65; p = 0.01; 95% CI = 1.12–2.43) where a one unit increase in DMN–SN rsFC resulted in a 65% increase in the probability of quitting the DT task (Figure 1b). 

### 3.5. Unique Contribution of Within and between Network rsFC on DT

A final Cox proportional hazard regression model tested the unique influence of within and between network rsFC on DT (Appendix A). Based on the results of previous analyses showing that the main effect of LECN and DMN-SN significantly predicted DT, both variables were entered simultaneously along with the group. The overall model was significant (χ^2^(3) = 12.64, p = 0.005), with both LECN (HR = 0.64; p = 0.008; 95% CI = 0.46–0.89) and DMN–SN (HR = 1.70; p = 0.02; 95% CI = 1.11–2.61) rsFC remaining significantly associated with DT. Interpretation of the odds ratios revealed that a one unit increase in LECN rsFC was associated with a 36.1% decrease in likelihood of quitting the DT task, while a one unit increase in DMN–SN rsFC was associated with a 70.2% increase in the likelihood of quitting the DT task. In other words, reduced LECN and greater DMN–SN rsFC predicted higher DT. 

### 3.6. Cocaine Use Severity, rsFC, and Distress Tolerance

Individuals in the CU group reported an average cocaine use frequency ratio of 0.21 ± 0.21 (i.e., 21% of past 30 days) and an average of 0.44 ± 0.51 grams of cocaine per using day in the 30 days prior to scan. Regression models tested the influence of each cocaine use severity variable on both DT and rsFC within the CU group (Appendix A). Neither the amount of cocaine use per using day nor cocaine use frequency were significantly associated with DT. Cocaine use frequency was not significantly associated with LECN or DMN–SN rsFC. However, amount of cocaine use per using day was significantly related to DMN–SN rsFC, such that a higher amount of cocaine use per using day was associated with greater DMN–SN rsFC (B = 1.15, Standard Error = 0.47, 95% CI = 0.18, 2.12, p = 0.02).

## 4. Discussion

Study findings support the hypothesis that resting-state functional connectivity (rsFC) within and between the executive control (ECN), default mode (DMN), and salience (SN) networks contribute to distress tolerance (DT). First, greater LECN and L.RAI rsFC, interpreted as greater LECN–SN relative to DMN–SN rsFC, are associated with greater DT. Such findings build upon previous work linking DT to activation in, and connectivity between, sub-regions of large-scale neural networks (e.g., [26]) by demonstrating that both the integrity of such networks (i.e., ECN in particular) as well as coordinated activity between specific networks (i.e., SN and DMN) are particularly important predictors of DT not only in CU individuals, but also healthy controls. As such, this study provides evidence that ECN-supported processes such as engagement in goal-oriented behavior and inhibitory control (e.g., [37,38]), are critical for DT. Perhaps more importantly, the findings also provide evidence that the allocation of attentional resources toward internally relevant stimuli (e.g., withdrawal symptoms) at the expense of engagement in externally oriented action, results in an inability to tolerate distress in favor of a longer-term goal [13,14,39]. Thus, it may be that mechanisms contributing to DT (and alternatively DT impairments in individuals with SUD) mimic negative reinforcement processes thought to occur in withdrawal-related relapse to substance use [10,13,14]. 

Second, the L.RAI effect on DT was driven by DMN–SN rsFC where greater DMN–SN rsFC was associated with increased likelihood of quitting the DT task (i.e., lower DT). This is in line with previous work demonstrating differences in L.RAI rsFC driven by DMN–SN rsFC between abstinent and satiated smokers [24]. Building upon what has been tested in previous research, this study additionally found that both LECN and DMN–SN rsFC were particularly important predictors of DT when combined into a single model, though DMN–SN rsFC appeared to have a stronger influence on DT when compared to LECN rsFC. As such, these findings are consistent with neurobiological models of SUD emphasizing the importance of both within and between network connectivity among the ECN, DMN, and SN in the maintenance of SUD and withdrawal-related processes [13,14,39]. This notion is further supported by the finding that cocaine use severity, particularly the amount of cocaine used per using day, is associated with stronger DMN–SN rsFC. This corresponds with the report of greater DMN-SN rsFC in relapsing heroin-dependent men when compared to those in early remission and healthy controls [25] and is also in line with research demonstrating associations between ECN rsFC and substance use severity and relapse risk in individuals with alcohol use disorder [22] and numerous studies demonstrating associations between substance use severity and neural structure and function (see [40] for review). 

Contrary to the study hypotheses, we did not find significant associations between R.RAI and DT. Methodological considerations made in the current study, particularly to examine large-scale neural networks rather than individual regions of interest in relation to DT may have contributed to discrepancies between the findings of the current study and previous ROI-based research. Though significant group differences in RECN–SN rsFC were found, the lack of association between R.RAI and DT was particularly surprising, given the importance of neural connectivity between the right middle frontal gyrus and ventromedial prefrontal cortex/subgenual ACC, the primary nodes of the RECN, SN, and DMN, in a previous study delineating task-based neural indices of DT [26]. Investigation of network-based connectivity supports the idea that cognitive and behavioral processes such as DT result via the dynamic interplay within and between large scale neural networks rather than the isolated activity of specific regions [41]. However, when utilizing a network-based approach, there may be significant associations between sub-regions within specific networks that are important for DT (e.g., right-lateralized sub-regions of the ECN and SN; [23]), which may not be detected using this network-based method. Thus, future studies combining ROI and network-based approaches may provide additional specificity regarding impairments in DT among individuals with SUD.

Additionally, we did not find significant associations between DT and either DMN or SN rsFC, though CU did demonstrate greater DMN rsFC compared to the HC group. Moreover, the interaction between group and rsFC on DT was not observed. Much of what we know concerning the neural basis of addiction-related processes have historically been established using task-based paradigms or by investigating neural connectivity in a state-dependent context such as acute withdrawal (see [13,14,42]). For example, the DMN is conceptualized as a task-negative neural network whose activity often decreases during task performance [15], though this network is additionally implicated in self-referential processing and linked to ruminative behavior in individuals with major depressive disorder [43]. In line with negative reinforcement conceptualizations of DT and aberrant DMN connectivity findings among individuals with SUD (e.g., [44,45]), it may be that heightened DMN connectivity during distress, when aversive internal signals (e.g., withdrawal symptoms) are particularly salient, is associated with lower DT. Such associations, in addition to more salient group differences in associations between connectivity and DT, may be best captured using a task/state-based connectivity approach. 

Several strengths of this study are notable. First, the CU sample was predominantly African American adults recruited from a large metropolitan area. Cocaine is one of the most commonly used illicit substances in the United States [46], and African American adults from urban areas are an understudied and underserved population [47]. It remains unknown as to the extent to which the current study findings generalize to more racially and gender diverse samples, or those that primarily use substances other than cocaine, which is a fruitful area for future research. An additional methodological strength of this study was the use of a novel analytical approach to operationalize DT. Previous DT research predominantly operationalizes DT as a binary (i.e., did the participant quit the task, yes or no) or continuous (i.e., seconds until quit time) variable separately across two different tasks. We were able to combine these approaches with a time-to-event analysis. 

A number of limitations are of note. First, we used pre-defined template maps of rsFC networks [36] rather than cohort-specific network indices (e.g., independent components analysis (ICA)). Though the use of template maps allows for a standardized approach to the identification of networks of interest, an ICA approach allows for a test of group differences in network configurations. Such questions were outside of the scope of the current study; however, future work may wish to examine the properties of addiction-related neural networks that differ between CU and HC groups and how these differences impact DT. Second, the repetition time used in this study undersampled confounding physiological signals such as cardiac and respiration. As such, these signals likely reduced the signal to noise ratio, with a concomitant loss of statistical power. Finally, the sample size may have limited the ability to detect significant yet smaller effects for integrated models testing the effect of substance use severity and rsFC on DT among individuals in the CU group in an integrated statistical model. As such, statistical models interpreted in this study were not corrected for multiple comparisons. Future work may wish to examine the interrelationships between such variables in a larger sample of individuals who use cocaine. 

## 5. Conclusions

The current study reports that connectivity within the ECN as well as connectivity between the SN and DMN are associated with DT, a prominent substance use relapse risk factor. Such findings suggest that allocation of attentional resources toward internally relevant stimuli at the expense of engagement in externally oriented actions results in DT impairments. Moreover, associations between cocaine use severity and DMN–SN rsFC suggest a potential mechanism through which substance use impacts DT (e.g., [9]). The existence of neural network functional connectivity indices of DT among non-treatment seeking adults sets the stage for a prospective test of the interactive relationship between rsFC, DT, and relapse within a treatment seeking sample.

## Figures and Tables

**Figure 1 jcm-08-02135-f001:**
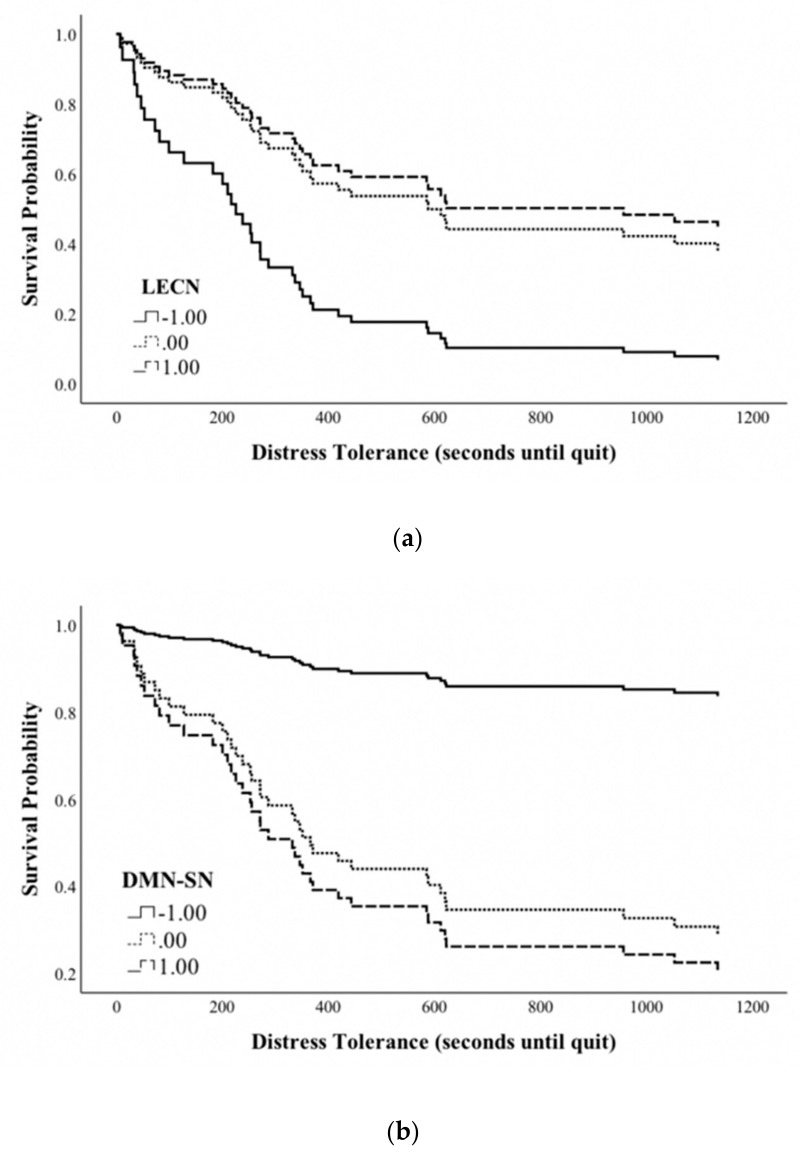
Probability of DT over time (seconds) for individuals with low (−1 SD), medium (mean), and high (+1 SD) (**a**) LECN and (**b**) DMN-SN rsFC. DT = distress tolerance; SD = standard deviation; LECN = left executive control network; DMN = default mode network; SN = salience network

**Table 1 jcm-08-02135-t001:** Participant characteristics.

	Total Sample (*n* = 57)	HC (*n* = 28)	CU (*n* = 29)	Statistic
Age *_(Mean, SD)_*	40.46 (8.44)	39.57 (8.90)	41.31 (8.04)	*t*(55) = −0.77
Gender _(*N*male, %)_	44 (77.20)	17 (60.70)	27 (93.10)	*χ^2^*(1) = 8.49 **
Race _(*N*Black/A.A. %)_	44 (77.20)	21 (75.00)	23 (79.30)	*χ^2^*(1) = 0.15
IQ *_(Mean, SD)_*	103.63 (12.36)	106.29 (12.75)	101.56 (11.88)	*t* (46) = 1.33
Anxiety Symptoms *_(Mean, SD)_*	2.88 (5.21)	1.72 (3.14)	4.04 (6.53)	*t*(54) = −1.69
Depression Symptoms *_(Mean, SD)_*	4.38 (6.46)	2.21 (3.56)	6.54 (7.91)	*t*(54) = −2.63 *
Within rsFC *_(Mean, SD)_*				
DMN	0.45 (0.05)	0.44 (0.04)	0.47 (0.03)	*t*(55) = −2.26 *
SN	0.41 (0.04)	0.41 (0.04)	0.41 (0.05)	*t*(55) = 0.26
RECN	1.23 (0.44)	1.30 (0.42)	1.17 (0.45)	*t*(55) = 1.13
LECN	0.75 (0.49)	0.89 (0.46)	0.62 (0.49)	*t*(55) = 2.13 *
Between rsFC *_(Mean, SD)_*				
L.RAI	−0.96 (0.39)	−1.04 (0.40)	−0.88 (0.36)	*t*(55) = −1.56
R.RAI	−0.74 (0.32)	−0.69 (0.30)	−0.79 (0.33)	*t*(55) = 1.26
DMN-SN	0.73 (0.10)	0.74 (0.09)	0.72 (0.11)	*t*(55) = 0.48
LECN-DMN	0.44 (0.18)	0.48 (0.13)	0.40 (0.22)	*t*(55) = 1.73
RECN-DMN	−0.07 (0.23)	−0.12 (0.24)	−0.02 (0.22)	*t*(55) = −1.63
LECN-SN	0.00 (0.23)	−0.06 (0.23)	0.06 (0.22)	*t*(55) = −2.05 *
RECN-SN	0.20 (0.23)	0.27 (0.19)	0.14 (0.24)	*t*(55) = 2.18 *

HC = healthy control; CU = cocaine use; SD = standard deviation; rsFC = resting state functional connectivity; L = left: R = right: RAI = resource allocation index: ECN = executive control network: SN = salience network: DMN = default mode network: * *p* < 0.05, ** *p* < 0.01.

**Table 2 jcm-08-02135-t002:** Distress tolerance task effects.

	Total Sample (*n* = 57)	HC (*n* = 28)	CU (*n* = 29)	Statistic
Task Order _(MT first *N*, %)_	25 (43.90)	13 (46.40)	12 (41.40)	*χ^2^*(1) = 0.15
**Task 1:**				
Performance ^†^ *_(Mean, SD)_*	0.19 (1.11)	0.14 (0.97)	0.25 (1.27)	*F*(1,46) = 0.11
Motivation *_(Mean, SD)_*	8.38 (1.83)	8.45 (1.84)	8.31 (1.85	*F*(1,54) = 0.06
Distress *_(Mean, SD)_*				^a^*F*(1,53) = 31.60 ***
Pre-Easy	10.83 (16.66)	8.25 (14.39)	13.32 (18.50)	*^b^F*(1,53) = 1.31
Pre-DT	26.36 (25.47)	22.73 (22.80)	29.87 (27.76)	*^c^F*(1,53) = 0.21
**Task 2:**				
Performance^†^ *_(Mean, SD)_*	−0.20 (0.83)	−0.13 (0.82)	−0.27 (0.85)	*F*(1,44) = 0.27
Motivation *_(Mean, SD)_*	8.25 (1.70)	7.96 (1.79)	8.54 (1.60)	*F*(1,52) = 1.48
Distress *_(Mean, SD)_*				^a^*F*(1,52) = 18.76 ***
Pre-Easy	18.94 (24.26)	14.84 (21.85)	22.90 (26.14)	*^b^F*(1,52) = 1.96
Pre-DT	30.02 (27.64)	24.44 (26.45)	35.22 (28.16)	*^c^F*(1,52) = 0.19

All models include task order as covariate; ^a^, Main effect of time. ^b^, Main effect of group. ^c^, Group × time interaction; ^†^, indicates variable calculated using standardized values; HC = healthy control; CU = cocaine use; DT = distress tolerance; *** *p* < 0.001.

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
