# Peer review of "Triple Network Resting State Connectivity Predicts Distress Tolerance and Is Associated with Cocaine Use"

_jcm, 2019, doi:10.3390/jcm8122135_

Round 1

Reviewer 1 Report

The authors study the relationship between distress tolerance and BOLD-based resting state functional connectivity in three brain networks in a sample of 29 adult cocaine users and 28 matched controls. The authors formulate multiple sensible hypotheses, based on literature and found evidence in support of multiple of these hypotheses in this data set. The study seems very well designed and seems to adequately control for potential confounds. The article is written really well and should allow others to replicate the findings. I would recommend though, that the authors consider to reduce the amount of jargon in the text to make the article more easily accessible to a broader audience. Below some additional comments.

Where the authors describe data, like first at the end of 2.1 but also throughout the manuscript I would strongly recommend to try to be unambiguous about whether you report mean or median and what metric you report for dispersion.

For Figure 1, please consider not using the colors red and green since some readers may have trouble telling these colors apart. I also recommend to make the lines thicker and increase font sizes for better readability. It’s not quite clear to me, why the abscissa labeling would go up to 1.2 million seconds (~2 weeks). From the text it seems the maximum might have to be 1200 seconds (from 10 min task length * 60 s => 600 s * 2 tasks => 1200 s).

In 3.5., where it says “A final Cox proportional hazard regression model was conducted…”, I believe you may have meant to write something like “A … model was fit…” or “A … regression analysis was conducted…”.

With a rather long TR of 2 seconds, there may be aliasing of signals of higher frequencies into low frequencies that are relevant for resting state analysis. Similarly, did you correct for pulsatory and respiratory artifacts? I think it would be good to discuss these topics as limitations of this study, but more importantly, these noise sources may have decreased the signal to noise ratio of the data that went into your analysis. This could have decreased the power of your statistical analysis, and could have been a contributing factor in why some of your predicted effects did not reach statistical significance.

Where the authors write “This is important to evaluate given the emphasis of large-scale network connectivity in neurobiological models of SUD”, please consider providing a suitable reference to support your statement.

Apologies, if I overlooked this in the text: Please make sure you explain which IRB approved your study protocol and where in the process subjects provided written, informed consent. This seems particularly good to mention here, since your study involved an element of deception. The “Instructions for Authors” for the journal seem to suggest the methods section as an appropriate location for this information.

Please, definitely make clear in the manuscript where the full data set can be downloaded, so that other researchers can replicate your findings.

Please include a statement about which kind of power analysis you performed to justify the number of recruited subjects.

Reviewer 2 Report

Reese and colleagues present a statistical analysis of resting-state connectivity to predict distress tolerance, as measured by two different cognitive tasks, and cocaine use. I applaud the authors on a clear and well-written manuscript. They first find that connectivity within portions of the default mode network and between portions of the executive control network and salience network discriminate cocaine users from healthy controls. They also find that connectivity within portions of the executive control network and between the default mode network and salience network are predictive of distress tolerance, as measured by the probability of quitting the distress-inducing task over time. The study strengths and limitations are well defined based on the characteristics of the cohort. The study is an important contribution into understanding the mechanisms of resource allocation, as inferenced by network-level interactions, towards distress tolerance.

I have no major concerns and a few minor points below:

In the section on Task Data (lines 163-173), as written, it is unclear if a participant who quits early in task 1 is able to move on to task 2. The example scenario in which DT is calculated does not highlight whether a composite score of DT is computed between task 1 and task 2 if task 1 is quit early.

Why is laterality in the executive control network considered as a predictor if laterality in the DMN and SN are not considered? Would it not make for a more comprehensive analysis if there were unique cross-hemispheric interactions associated with DMN and SN?

There are a few different models that are constructed and assessed in this study. Are the statistical tests controlled for multiple comparisons? Whether or not multiple comparisons testing was used should be clarified throughout the text.
